# Analysis of Operational Parameters in Acid and Base Production Using an Electrodialysis with Bipolar Membranes Pilot Plant

**DOI:** 10.3390/membranes13020200

**Published:** 2023-02-06

**Authors:** Marta Herrero-Gonzalez, Julio López, Giovanni Virruso, Calogero Cassaro, Alessandro Tamburini, Andrea Cipollina, Jose Luis Cortina, Raquel Ibañez, Giorgio Micale

**Affiliations:** 1Departamento de Ingenierías Química y Biomolecular, Universidad de Cantabria, 39005 Santander, Cantabria, Spain; 2Dipartamento di Ingeniería, Università degli Studi di Palermo, 90128 Palermo, Italy; 3Chemical Engineering Department, Escola d’Enginyeria de Barcelona Est (EEBE), Universitat Politècnica de Catalunya (UPC)-Barcelona TECH, Campus Diagonal-Besòs, 08930 Barcelona, Cantabria, Spain; 4Barcelona Research Center for Multiscale Science and Engineering, Campus Diagonal-Besòs, 08930 Barcelona, Cantabria, Spain; 5ResourSEAs SrL, 90128 Palermo, Italy; 6CETaqua, Carretera d’Esplugues, 75, 08940 Cornellà de Llobregat, Barcelona, Spain

**Keywords:** brine, FuMa-Tech, sodium hydroxide, hydrochloric acid, circular economy, electrodialysis with bipolar membranes (EDBM)

## Abstract

In agreement with the Water Framework Directive, Circular Economy and European Union (EU) Green Deal packages, the EU-funded WATER-MINING project aims to validate next-generation water resource solutions at the pre-commercial demonstration scale in order to provide water management and recovery of valuable materials from alternative sources. In the framework of the WATER-MINING project, desalination brines from the Lampedusa (Italy) seawater reverse osmosis (SWRO) plant will be used to produce freshwater and recover valuable salts by integrating different technologies. In particular, electrodialysis with bipolar membranes (EDBM) will be used to produce chemicals (NaOH and HCl). A novel EDBM pilot plant (6.4 m^2^, FuMa-Tech) has been installed and operated. The performance of EDBM for single pass under different flowrates (2–8 L·min^−1^) for acid, base and saline channels, and two current densities (200 and 400 A·m^−2^), has been analyzed in terms of specific energy consumption (SEC) and current efficiency (CE). Results showed that by increasing the flowrates, generation of HCl and NaOH slightly increased. For example, ΔOH^−^ shifted from 0.76 to 0.79 mol·min^−1^ when the flowrate increased from 2 to 7.5 L·min^−1^ at 200 A·m^−2^. Moreover, SEC decreased (1.18–1.05 kWh·kg^−1^) while CE increased (87.0–93.4%), achieving minimum (1.02 kWh·kg^−1^) and maximum (99.4%) values, respectively, at 6 L·min^−1^.

## 1. Introduction

Nowadays, it is estimated that 4.0 billion people (c.a. 2/3rds of the world population) face severe water scarcity during at least part of the year [1]. Moreover, it is expected that water demand will increase by 50–80% over the next three decades due to population growth, urbanization, socioeconomic development, intensification of agriculture, quality deterioration of available water and climate change [2,3]. By 2050, near half of the global urban population will live in water-scarce regions [2]. High water scarcity levels appear in areas with: (i) high population density, (ii) irrigated agriculture, or (iii) very low natural water availability (arid areas) [1]. In addition, even though water scarcity can be addressed by improving the water-use efficiency, or via seawater desalination, groundwater exploitation, or by applying water disinfection and decontamination techniques, these management options will affect the natural environment, as they are chemically, energetically and operationally intensive [2,4]. Therefore, some actions must be taken to ensure a reliable water supply, such as (i) reducing pressure on limited blue water resources by using green water (e.g., rainwater) or by increasing the productivity in agriculture, (ii) applying emerging water-saving technologies, (iii) controlling population growth in water-scarce regions and (iv) mitigating climate change through energy efficiency and emissions abatement measures [1,2].

To date, water reuse and desalination technologies for either sea or brackish waters from saline aquifers and rivers are the only options that can secure reliable access to water in terms of quantity and quality [3,4,5,6]. It is estimated that by increasing the desalination capacity from 2.9 to 13.6 billion m^3^·month^−1^ and water reuse from 1.6 to 4.0 billion m^3^·month^−1^, it can be possible to reduce the percentage under severe water scarcity to 14% [3,7]. Reverse osmosis (RO) accounts for 70% of desalination plants. In comparison, the thermal-based processes (multi-stage flash distillation and multi-effect distillation) account for 25%, nanofiltration and electrodialysis being the remaining percentage [7]. The conventional thermal desalination processes are inefficient due to their large use of energy and high capital and operational costs (7.7–70.0 kWh·m^−3^) [4,7]. During the last decades, the continuous improvement of seawater desalination using RO, in terms of energy recovery and more robust membranes, has made it possible to reduce the energy consumption from 10 kWh·m^−3^ in the 1980s to below 4.0 kWh·m^−3^ of produced water [4]. In addition, from a theoretical point of view, energy consumption is expected to be decreased to 1.29 kWh·m^−3^ at a 75% recovery [4]. Nevertheless, major challenges of seawater RO desalination still remain, such as: (i) fouling, (ii) concentration polarization, (iii) low water recovery (<55%) and (iv) the need to remove low molecular weight contaminants (e.g., boron) [4].

In addition, there are some environmental concerns in relation to seawater RO desalination plants, such as: (i) the management of brines, (ii) emission of air pollutants and greenhouse gases (c.a. 1.4–1.8 kg CO_2_·m^−3^ of produced water), (iii) entrainment of planktonic organisms and juvenile-stage fish in the sea water uptake systems, and (iv) the release of chemicals to the environment (e.g., anti-scalants) [5]. Concerning the brine discharge, it has been reported that brine salinity (c.a. 65–85 g·L^−1^) is the main factor that impacts the environment [8]. Moreover, some chemicals such as ClO_2_(g), NaOCl, FeCl_3_, AlCl_3_, H_2_SO_4_, HCl and NaHSO_3_ are usually dosed in the pre-treatment stage to avoid algal growth, reduce corrosion, inhibit scaling and adjust the pH and chlorinating water [8,9]. One example of the harm caused by the brine discharge is the meadows regression of *Posidonia oceanica* in the Mediterranean area [10,11]. Several methods to deal with brine management, such as evaporation, membrane distillation, well injection and dilution with a wastewater treatment plant or seawater can be applied [9,12]. In order to reduce the volume of brine and its disposal cost, zero liquid discharge (ZLD) and near-ZLD approaches can be applied [13,14,15].

Within this context, the WATER-MINING project aims to ensure access to water by exploring alternative water sources through the development of innovative solutions that are being tested with urban and industrial wastewater and seawater desalination [16]. Case Study 1, placed at the seawater reverse osmosis (SWRO) of Lampedusa (Italy), evaluates the performance of innovative technologies to avoid brine discharge by their valorization into chemicals or freshwater. The treatment chain (Figure 1), with an inlet flow capacity of 2.2 m^3^·h^−1^, is composed of a nanofiltration unit treating seawater, which splits the stream in two. The permeate (mainly containing NaCl and KCl) will be sent to a multiple-effect distillation unit (employing waste heat) to produce high-quality water. The remaining brine, almost NaCl-saturated, will be treated in a thermal crystallizer to obtain high-quality water and NaCl(s). On the other hand, the nanofiltration concentrate (rich in Mg and Ca) will be sent to a selective Mg and Ca crystallizer to precipitate them as hydroxides [17]. The remaining solution (Na_2_SO_4_ and NaCl mixture) will be treated in the Eutectic Freeze Crystallization unit to precipitate Na_2_SO_4_(s). Finally, the residual brine (mainly NaCl) will be fed to the electrodialysis with bipolar membranes (EDBM) unit to produce NaOH and HCl that can be reused internally in the process [18]. The proposed treatment scheme would allow a more competitive desalination process due to the higher freshwater production and water recovery, the lower energy consumption and the production of high-purity salts (NaCl(s) and Na_2_SO_4_(s)) and chemicals (Mg(OH)_2_(s) Ca(OH)_2_(s), securing sustainable access to Mg (listed as a Critical Raw Material) within the European Union. Moreover, the volume of brine discharged to the environment will be reduced. 

EDBM is an electro-membrane process that uses a repeating sequence of anion exchange, cation exchange and bipolar membranes (AEM, CEM and BPM, respectively) that can convert salts into acids and bases under the application of an external current field [19,20,21]. The use of EDBM has been widely studied at a laboratory scale for the production of acid and bases from brines [22,23,24,25,26,27]. For instance, Yang et al. [23] treated RO brine from a power plant using a set-up provided by Shandong Tianwei Membrane Technology Co., Ltd. (88 cm^2^). The stack was equipped with AEM and CEM from Qianqiu Environmental Protection & Water Treatment Corporation (China), while the BPM was from FuMa-Tech GmbH (Germany). In order to avoid scaling, the RO brine was previously treated with NaOH and CO_2_(g) to precipitate Mg(OH)_2_ (>99.9%) and CaCO_3_(s) (>99%). By applying a current density of 570 A·m^−2^ working in batch configuration, the authors were able to produce 0.65 mol·L^−1^ H^+^ with a SEC of 9.0 kWh·kg^−1^ HCl. Davis et al. [24] produced acids and bases from NaCl solutions (50 mmol·L^−1^ to 400 mmol·L^−1^) using Neosepta membranes (ACM, CMX and BP−1) in a lab-scale unit (PC-Cell EDQ380, 380 cm^2^). Tests carried out in a single pass showed that it was possible to achieve a 75% salt conversion working at an inlet of 192 mmol·L^−1^ NaCl and 192 A·m^−2^. Under these conditions, authors achieved a SEC of 3.65 kWh·kg^−1^ HCl and a current efficiency higher than 80%. Reig et al. [25] integrated nanofiltration, selective precipitation and EDBM for treating seawater RO brines from El Prat de Llobregat (Spain). The permeate of the nanofiltration (52 g·L^−1^ NaCl, 0.76 g·L^−1^ Mg and 0.42 g·L^−1^ Ca) was treated with Na_2_CO_3_ and NaOH to precipitate CaCO_3_(s) (removal of 96%) and Mg(OH)_2_(s) (removal of 99%). This would allow having residual amounts of Mg (<10 mg·L^−1^) and Ca (<20 mg·L^−1^) prior to the treatment with EDBM. Experiments carried out in batch configuration using EDBM at 450 A·m^−2^ showed the possibility of producing 0.88 mol·L^−1^ of HCl and 0.71 mol·L^−1^ of NaOH with a current efficiency of 77% and a SEC of 2.58 kWh·kg^−1^ of NaOH. Recently, Herrero-González et al. [26] tested the production of HCl and NaOH from 1 mol·L^−1^ of NaCl solutions in a PC-Cell lab-unit (100 cm^2^) using Ralex (CM-PP and AM-PP) and Fumasep (BPM) membranes. Working in a constant and variable (solar photovoltaic) current intensity mode at 1000 A·m^−2^ and a volume salt to acid (or base) ratio of 20, they were able to achieve 3.2 mol·L^−1^ of HCl and 3.6 mol·L^−1^ of NaOH with a SEC value of 41 kWh·kg^−1^ of HCl after 40 h of operation. 

Although EDBM has been validated at a laboratory scale (technology readiness level (TRL) 4) and even in a real environment (TRL 5), in the open literature there is still a lack of knowledge about the operation of EDBM plans with a higher TRL (TRL 6, i.e., technology demonstrated in a real environment) [28]. To the authors knowledge, there is only one paper that evaluates the performance of EDBM at the pilot plant scale. Gazigil et al. [27] fed the RO concentrate to treat wastewater in an EDBM pilot plant. The module provided by FuMa-Tech (EDBM FT-ED 100) has an active area of 100 cm^2^ with a feed flowrate of 180 L·h^−1^ (10 triplets). The system operated under batch configuration at 25 V, but once the brine was exhausted, it was replaced with the fresh one keeping the solutions for acid and base. After performing six cycles of operation, concentrations of 1.44% and 2% of acids and bases, respectively, were reached. No data on specific energy consumption or current efficiency was provided. 

Very recently, the authors have presented the production of HCl and NaOH from NaCl brines by means of EDBM in a novel pilot plant [29]. This EDBM unit is configured with a total membrane area of 19.2 m^2^, more than 16 times larger than that reported in the literature for the same application. The pilot unit was operated in three operation modes (batch, feed and bleed and fed-batch) and two current densities (200 and 400 A·m^−2^). At 200 A·m^−2^, the lowest SEC (1.4 kWh·kg^−1^) and highest CE (80%) were obtained operating in batch mode. However, at 400 A·m^−2^, the feed and bleed operation mode presented lower SEC (2.1 kWh·kg^−1^) and higher specific production (SP) (1.1 ton·y^−1^·m^−2^).

This work evaluates the performance of a novel EDBM pilot plant (FT-ED1600-3, FuMa-Tech), one of the largest stacks that has been tested in the literature. In particular, its total membrane area is much larger (i.e., more than 16 times larger) than those reported so far for the production of HCl and NaOH aqueous solutions, starting from NaCl brines. The aim of this work was to determine the effect of flowrates (acid, base, salt and ERS), initial concentrations in the compartments and the current density in a one-pass configuration using single-salt (NaCl) solutions on the increase of H^+^ and OH^−^ concentrations, SEC_EDBM_ and CE_EDBM_. Additionally, the contribution of pump consumption was considered within the SEC_Total_.

## 2. Materials and Methods

### 2.1. Experimental Set-Up

The EDBM pilot plant (Figure 2) is composed by two skid-mounted units: (i) the pumping, monitoring and control station, and (ii) the EDBM stack. The former contains the pumps to feed the solution to the stack and the instrumentation devices to monitor and control the main variables involved in the process, while the latter is composed using an EDBM FT-ED1600-3 module provided by FuMa-Tech BWT GmbH (Bietigheim-Bissingen, Germany).

The EDBM stack was equipped with homogeneous FUMASEP^®^ FAB-PK, FKB-PK and FBM-PK as AEM, CEM and BPM respectively, also supplied by FuMa-Tech BWT GmbH (Bietigheim-Bissingen, Germany). Their main properties are collected in Table 1 [30]. Each channel is provided with net-spacers 350 µm thick and made of PP-overlapped wires (FuMa-Tech FT-ED-1600-3). The stack is configured with 2 sets of 20 triplets (repeating units of anionic, cationic and bipolar membranes), which sums an area of 6.4 m^2^ for each type of membrane; this means a total membrane area of 19.2 m^2^.

The three main line of process (acid, base and salt) were equipped with 2 tanks (IBC) of 1 m^3^ each. These tanks together with a complex system of piping allow the operation of the pilot plant in different process configuration (batch, continuous, and feed and bleed mode) [29]. In addition, a cylindrical tank of 0.125 m^3^ is used for the electrode rinse solution (ERS), which is continuously recirculated. 

A DC drive (AF02, Giussani SRL, Milano, Italy) capable of providing 17.5 kW is installed at the pilot plant to provide the electrical energy to power the EDBM unit. 

The continuous sampling of the process variables (inlet and outlet pH, flowrate and conductivity in the four streamlines, as well as stack current and voltage) was obtained using a CompactDAQ model NI cDAQ-9179, provided by National Instruments (NI, Austin, TX, USA). Moreover, a LabVIEW (NI, Austin, TX, USA) interface was built to show the trend of the main variables and to control the system.

### 2.2. Experiments Design

A total of three sets of experimental runs have been carried out in the EDBM pilot plant. Table 2 depicts a summary of experimental conditions adjusted. Note that the pilot has been operated at 200 and 400 A·m^−2^ because: (i) these values are representative of a wide range of current densities, (ii) lower than the supplier-recommended upper limit of 500 A·m^−2^ and (iii) lower than the limiting current density estimated using a validated process model [31]. 

In the first set, the initial concentrations have been fixed to be the initial values of a batch experiment (ca. 0.1 mol·L^−1^ HCl and NaOH and 1 mol·L^−1^ NaCl), and the flowrates of acid, base and saline channels were varied from 2 to 8 L·min^−1^, while the one of ERS remained constant (20 L·min^−1^). Only the experiment T-200-F was performed at 8 L·min^−1^, whereas the other tests were performed at 7.5 L·min^−1^. This was due to the high inlet pressure (>2 bar) that was observed when working at 8 L·min^−1^, which caused external leakage. By working at 7.5 L·min^−1^, the pressure remained below two bar, which avoided the presence of an external leakage. Similarly, the second set varied the initial concentrations of the tests, which were those close to the end of a batch experiment (ca. 0.7 mol·L^−1^ HCl and NaOH and 0.5 mol·L^−1^ NaCl). In this case, the flowrates were varied within the same range. Finally, the third set was focused on studying the effect of ERS flowrates (from 5 to 30 L·min^−1^) with the initial concentration of the first set and by keeping the acid, base and saline flowrates at 6 L·min^−1^. In all the sets, experiments were performed at 200 and 400 A·m^−2^.

### 2.3. Experimental Procedure

Firstly, the acid, base and saline tanks were filled with 600 L and the ERS tank was filled with 125 L of the corresponding solutions for each test (Table 2). The main purpose of the experiments was to evaluate the performance of the EDBM stack operating on the once-through (i.e., single-pass) operation mode. This configuration guarantees that the osmotic phenomena lead to a negligible variation of the volumes of the three solutions. Then, the system was powered up by fixing the initial flowrates and current density. After 15 min of stabilization, samples at the inlet and outlet of the stack for the four channels were collected via purposely installed valves, thus characterizing a single pass through the stack. After collecting the samples, they were stored at 4 °C. Once these tasks were completed, the flowrates were changed, starting the procedure from the stabilization.

In addition to the samples, both manual and automatic (using LabVIEW software) records of current, voltage, flowrates, pressure and temperature were carried out.

The pH and conductivity of the samples was measured offline using a multi-meter (Xylem Analytic Germany FmbH D-82362, Weilheim, Germany). In order to measure the concentrations of acid and bases, titration was executed twice per sample using a standard HCl 0.10 mol·L^−1^ and Na_2_CO_3_ 0.05 mol·L^−1^ solutions, and methyl orange as pH indicator.

The accuracy of sample measurement was evaluated through error propagation theory, considering the error of equipment, deviation between the titrations performed and the variability in the recorded values in LabVIEW. Thus, the possible deviations in the results reported are included as error bars in the graphs.

### 2.4. Calculation of Performance Parameters

Specific energy consumption (SEC_EDBM_) and current efficiency (CE_EDBM_) have been calculated for the evaluation of the performance of the EDBM process.

SEC_EDBM_ is an indicator of the energy consumed by the EDBM in the production of 1 kg of product. SEC_EDBM_ (kWh·kg^−1^) is calculated according to Equation (1):(1)$${\mathrm{SEC}}_{\mathrm{EDBM}}=\frac{I\cdot V}{\left(C_{\mathrm{out}}\cdot Q_{\mathrm{out}}-C_{\mathrm{in}}\cdot Q_{\mathrm{in}}\right)\cdot \mathrm{MW}}.\;$$
where I is the current (A); V is the voltage (V); C_in_ and C_out_ are the inlet and outlet concentrations in the channel (mol·L^−1^); Q_in_ and Q_out_ are the inlet and outlet flowrates to the stack (L·h^−1^); and MW the molecular weight (g·mol^−1^).

CE_EDBM_ measures the effectiveness of ion transportation across ion exchange membranes, thus the product generation, for a given applied current. CE_EDBM_ is calculated using the ratio between the equivalent charge transported per unit of time and the electric current (Equation (2)):(2)$${\mathrm{CE}}_{\mathrm{EDBM}}\;\left(\%\right)=\frac{\left(C_{\mathrm{out}}\cdot Q_{\mathrm{out}}-C_{\mathrm{in}}\cdot Q_{\mathrm{in}}\right)\cdot z\cdot F}{n\cdot I}\cdot 100$$
where C_in_ and C_out_ are the inlet and outlet concentrations in the channel (mol·L^−1^); Q_in_ and Q_out_ are the inlet and outlet flowrates to the stack (L·s^−1^); z is the ion valence; F is the Faraday constant (96485 C·mol^−1^); n is the number of membrane triplets within the stack (20); and I is the current (A).

Both SEC_EDBM_ and CE_EDBM_ can be referred either to NaOH or HCl, as they are simultaneously produced. However, in this work, SEC_EDBM_ and CE_EDBM_ are referred to NaOH as it is the product of higher economic market value [32]. Therefore, all the energy consumption of the EDBM is attributed to NaOH, whereas HCl is considered as a by-product.

Additionally, given that different flowrates in the channels are compared, the energy consumption related to the pumping of all the solutions through the different channels (P_cons_, W) was determined as the sum of pumping energy in the saline, acid, alkaline and ERS channels, according to Equation (3) [33].
(3)$$P_{\mathrm{cons}}=\overset{4}{\underset{i=1}{\sum }}\frac{Q_{\mathrm{in},i}\cdot (P_{\mathrm{in},i}-P_{\mathrm{out},i})}{\eta }$$
where Q_in,i_ is the inlet flowrate to the stack for channel i (m^3^·s^−1^); P_in,i_ and P_out,i_ are the inlet and outlet pressures to the stack for channel i (Pa); and η is the pump efficiency (70%).

The specific energy consumption associated to pumping (SEC_Pump_), referred to the kg of NaOH, can be calculated according to (Equation (4)):(4)$${\mathrm{SEC}}_{\mathrm{Pump}}=\frac{P_{\mathrm{cons}}}{\left(C_{\mathrm{out}}\cdot Q_{\mathrm{out}}-C_{\mathrm{in}}\cdot Q_{\mathrm{in}}\right)\cdot \mathrm{MW}}$$
where P_cons_ is the energy consumed by the pumping system (W); C_in_ and C_out_ are the inlet and outlet concentrations in the channel (mol·L^−1^); Q_in_ and Q_out_ are the inlet and outlet flowrates to the stack (L·h^−1^); and MW the molecular weight (g·mol^−1^).

The total specific energy consumption (SEC_Total_) is the sum of the SEC_EDBM_ and the SEC_Pump_ (Equation (5)):(5)$${\mathrm{SEC}}_{\mathrm{Total}}={\mathrm{SEC}}_{\mathrm{EDBM}}+{\mathrm{SEC}}_{\mathrm{Pump}}$$

## 3. Results and Discussion

### 3.1. Effect of Acid, Base and Saline Flowrates on the Performance of the EDBM Unit

The variation in concentration profiles (Figure 3) is equivalent in shape for the four cases in which the flowrates of the acid, base and saline channels were manipulated (T-200-I, T-400-I, T-200-F and T-400-F). In the four cases, both products, NaOH and HCl, also presented similar profiles. However, acid concentrations were slightly lower than for the base due to the higher mobility of protons that migrated to other compartments (saline or ERS) and to the non-ideality (permselectivity) of the AEMs.

The reduction of the concentration with the increasing flowrate was not linear but followed a trend inversely proportional to the inlet flowrate. As expected, the highest variation of the concentration was obtained at the lowest flowrate (2 L·min^−1^) since the residence time in the stack was increased. Considering the flow path (454 mm × 350 μm), the velocity of the fluid inside the channel was moved from 0.88 cm·s^−1^ (at 2 L·min^−1^) to 3.52 cm·s^−1^ (at 8 L·min^−1^). By considering the free volume of the spacer (47.25 cm^3^), the residence time of the stack was moved from 53.6 s to 13.4 s when the flowrate was increased from 2 to 8 L·min^−1^.

Higher variations of concentration were obtained when initial concentrations were employed (T-200-I and T-400-I) and when compared with tests performed at final concentrations (T-200-F and T-400-F). In the latter case, the larger concentrations achieved and the relevant pH gradient across the bipolar membranes generate a voltage which must be overcome to produce further chemicals. Thus, operating at the same current density intrinsically leads to a lower inlet–outlet concentration variation. Moreover, higher inlet concentrations favored the appearance of non-ideal phenomena (such as osmosis, electro-osmosis and diffusion) that worsen the EDBM performance.

On the other hand, working at a higher current density (400 A·m^−2^) increased the final acid and base concentrations and, therefore, the variation of concentrations. This is related to the presence of the electric field, which accelerates the kinetics of water dissociation at the bipolar membrane junction, as stated by the second Wien effect for under-limiting current densities [34]. In particular, close to the double concentration variation was obtained by shifting the current density from 200 to 400 A·m^−2^ (from T-200-I to T-400-I or from T-200-F to T-400-F).

As depicted in Figure 4, SEC_EDBM_ slightly decreased when the flowrate was increased in the four cases (T-200-I, T-400-I, T-200-F, and T-400-F). However, within the same experiment, there were no significant changes in the SEC_EDBM_. Since the energy supplied was approximately the same, the SEC_EDBM_ variations were due to the production, which, in turn, was dependent on the flowrate and the variation in concentration. Despite the fact that the variation in concentration was reduced, the production was slightly enhanced for higher flowrates, therefore, the SEC_EDBM_ was reduced.

Doubling the current density in the tests with initial conditions (T-200-I and T-400-I) made the SEC_EDBM_ 1.6–1.8 times larger, while in the tests with final conditions (T-200-F and T-400-F) the SEC_EDBM_ increased only 1.2–1.3 times. Surprisingly, the SEC_EDBM_ followed a quite horizonal tendency with the flowrate, except for the case of T-400-F. In this test, the SEC_EDBM_ reached values up to 3.2 kWh·kg^−1^ NaOH and followed a decreasing trend as the flowrate increased. This was related to the fact that the salt at the outlet of the stack was depleted (conductivity value of 3.8 mS·cm^−1^, see Figure A1), which caused: (i) an increase in the compartment resistance and therefore, an increase in voltage, but also (ii) a limitation of the water splitting phenomena due to the absence of counter-ions in the salt compartment that can compensate the charge balance.

On the other hand, for the tests with the same applied current density, the SEC_EDBM_ was 1.8 times higher for T-200-F compared to T-200-I, and 1.2–1.4 times higher for T-400-F compared to T-400-I. The variations were more remarkable in the tests with initial conditions (T-200-I and T-400-I) due to the contribution of the ohmic resistance in the voltage. The stack resistance was higher when the concentrations in the stack were lower (i.e., initial conditions, I), so the energy input was increased not only via a higher current but also via the voltage (composed by the electrode overpotential and the ohmic contribution). At the same time, the voltage was increased by the current and the resistance.

On the other hand, according to Figure 5, a small increase in CE_EDBM_ was observed as the flowrate increased, which was due to an increase in productivity.

For the same applied current densities (T-200-I and T-200-F, or T-400-I and T-400-F) the tests with the initial conditions had better CE_EDBM_ (1.8–2.0 times higher for T-200-I compared to T-200-F, and 1.3–1.4 times higher for T-400-I compared to T-400-F). This fact is due to the higher variations in the concentration obtained in the test with the initial conditions than those with the final conditions; that is, there was not so much of a loss of energy (current).

Looking at the CE_EDBM_, when the current density was increased, a reduction of 20% was observed in the tests with initial conditions, while only 10% differences were observed in the test at the final conditions.

### 3.2. Effect of ERS Flowrate on the Performance of the EDBM Unit

In this section, results from test T-200-I-ERS and T-400-I-ERS, in which the ERS flowrate was manipulated, are presented. Data related to the variation of electrical conductivities is reported in Figure A2.

The performance in the variation of concentration profiles (Figure 6) was different between the T-200-I-ERS (Figure 6a) and the T-400-I-ERS (Figure 6b) tests. For T-200-I-ERS, the acid concentration slightly decreased when increasing the flowrate, while the base concentration slightly increased. However, the changes throughout the test were small, for both products (<0.02 mol·L^−1^). In T-400-I-ERS, a maximum in the variation of concentration was observed for both products (although the acid concentration was somewhat lower due to the mobility and migration of protons) for the flowrate of 20 L·min^−1^, which was the flowrate recommended by the supplier for the ERS channel. In this case, the changes throughout the test were very small (<0.02 mol·L^−1^).

The final concentration and, therefore, the variation of concentration increased along with the current density. Almost twice the amount of concentration variation was achieved (for both NaOH and HCl) when the current density was doubled (from 200 A·m^−2^ in T-200-I-ERS to 400 A·m^−2^ in T-400-I-ERS).

Both the SEC_EDBM_ (Figure 7) and the CE_EDBM_ (Figure 8) were slightly affected by the variations in concentration obtained.

A minimum in the SEC_EDBM_ was observed for the flowrate of 20 L·min^−1^ (nominal flowrate) for both T-200-I-ERS and T-400-I-ERS.

The SEC_EDBM_ increased 1.3–1.5 times when the current density was doubled. Since the variation in concentration was doubled by changing the current density, the increase in SEC_EDBM_ was due to the higher power consumption, and, in particular, to the voltage associated with the ohmic behavior (in turn, it is dependent on the current).

Similar behavior was observed for the T-200-I-ERS and the T-400-I-ERS tests (Figure 8), where a maximum in the CE_EDBM_ was observed for the flowrate of 20 L·min^−1^ (nominal flowrate). The changes in CE_EDBM_ throughout the test were <13% for T-200-I-ERS and <5% for T-400-I-ERS. The changes in the CE_EDBM_ by doubling the current density were minimal (<5%) since twice the variation of the concentration was achieved.

### 3.3. Evaluation of Pumping Contribution to SEC_Total_

In this section, results from the calculation of the SEC_Pump_ and SEC_Total_ are presented.

Increasing the flowrate by definition increases the pumps consumption (Equation (3)), and therefore the SEC_Pump_, as seen in Figure 9 and Figure 10. Although the SEC_Pump_ presented values two orders of magnitude lower than the SEC_EDBM_, it may have an impact on the estimation of the SEC_Total_ given that the SEC_EDBM_ presents values with small variations, but with a growing trend regarding the increase in flowrate. In particular, the contribution of the SEC_Pump_ to SEC_Total_ in the tests employing 200 A·m^−2^ current density (T-200-I and T-200-F) was in the range of 1.3–6.9%, while in the 400 A·m^−2^ current density tests (T-400-I and T-400-F) it was in the range of 0.5–2.3%.

For the tests in which the acid, base and saline channel flowrates have been manipulated (T-200-I, T-200-F, T-400-I and T-400-F), different trends for the SEC_Total_ were observed depending on the current density employed. Due to the similar values obtained with the tests T-200-I and T-200-F (Figure 9a,c), an Analysis of Variance (ANOVA) was performed to evaluate if the values were statistically different (F > F_cr_). Accordingly, the analysis of the values obtained for both tests reported that the SEC values were statistically different (10.1 > 5.9 for T-200-I and 6.1 > 5.9 for T-200-F). Therefore, it can be observed that a minimum value for SEC_Total_ was obtained for the tests T-200-I and T-200-F (Figure 9a,c) between flowrates of 4 and 6 L·min^−1^, which may correspond to the nominal flowrate for these compartments (5 L min^−1^). On the other hand, for tests T-400-I and T-400-F (Figure 9b,d), there was a decreasing trend for SEC_Total_, that is, the lowest value of SEC_Total_ was given by the highest flowrate.

For the tests in which the ERS channel flowrate has been manipulated (T-200-I-ERS and T-400-I-ERS), different trends for the SEC_Total_ were reported, with a decreasing trend for the increase in the flowrate in T-200-I-ERS (Figure 10a), while in T-400-I-ERS (Figure 10b) a minimum value for SEC_Total_ was obtained in the nominal flowrate (20 L·min^−1^). Contributions of 3.2–6.4% and 1.2–2.3% for tests T-200-I-ERS and T-400-I-ERS were respectively obtained.

## 4. Conclusions

In this work, the effect of flowrates of acid, base, salt and ERS solutions, and the current density on the performance of EDBM in terms of H^+^ and OH^−^ generation, SEC_EDBM_ and CE_EDBM_ have been evaluated.

In relation to the effect of acid, base and salt flowrates, it was observed that the values tested reported a higher increase in the concentrations of acid and base in one single pass. It was also observed that the concentrations of base were slightly higher than the ones of acid due to the higher H^+^ diffusion and the non-ideality of the AEMs. Regarding the concentrations of the solutions at the beginning of the experiment, the higher their values, the higher the H^+^ and OH^−^ concentrations. This was related to a higher contribution of non-ideal phenomena when working with higher concentrations. As expected, a higher current density enhanced the production of H^+^ and OH^−^ in agreement with the second Wien effect. It was observed that the best values for SEC_EDBM_ (1.02–1.18 kWh·kg^−1^ NaOH) and CE_EDBM_ (ca. 100%) were attained working at higher saline concentrations and low current densities.

On the contrary, the ERS flowrate showed little effect on the H^+^ and OH^−^ concentrations, as well as on SEC_EDBM_ and CE_EDBM_. Nevertheless, a minimum value of SEC_EDBM_ was observed when working at 20 L·min^−1^ ERS (nominal operating conditions), independently of the current density applied. 

Finally, the contribution of the pumping needs to the SEC_Total_ was determined. As expected in the electro-membrane processes, the energy consumed related to the EDBM represented the majority of the contribution to the SEC_Total_. In all the scenarios, the pumping needs remained below 7%.

## Figures and Tables

**Figure 1 membranes-13-00200-f001:**
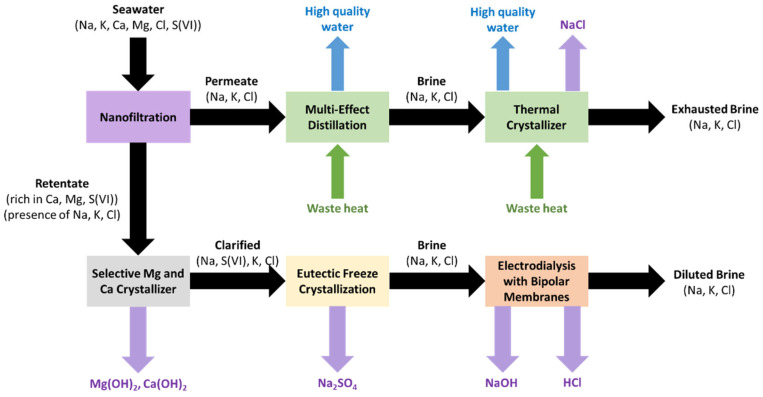
Scheme of the treatment chain evaluated in Case Study 1 (Lampedusa, Italy) within the WATER-MINING project (adapted from Culcasi et al. [18]).

**Figure 2 membranes-13-00200-f002:**
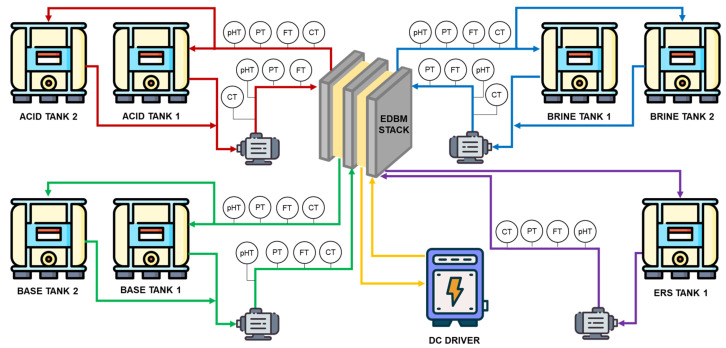
Simplified flowsheet of the EDBM pilot plant (adapted from [29]).

**Figure 3 membranes-13-00200-f003:**
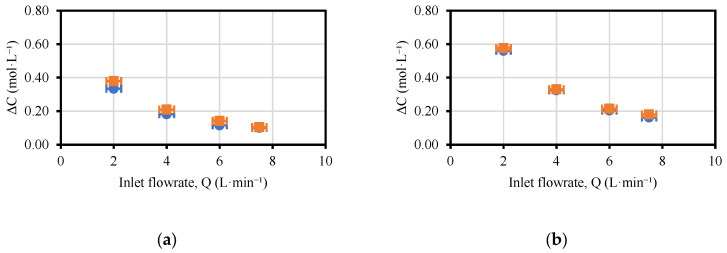
Values of concentration variation versus feed flowrate: (**a**) T-200-I; (**b**) T-400-I; (**c**) T-200-F; (**d**) T-400-F. The associated errors to the variation in the acid and base concentrations (vertical error bars), estimated via acid–base titrations, are below 3% and cannot be visualized in practice in the figure.

**Figure 4 membranes-13-00200-f004:**
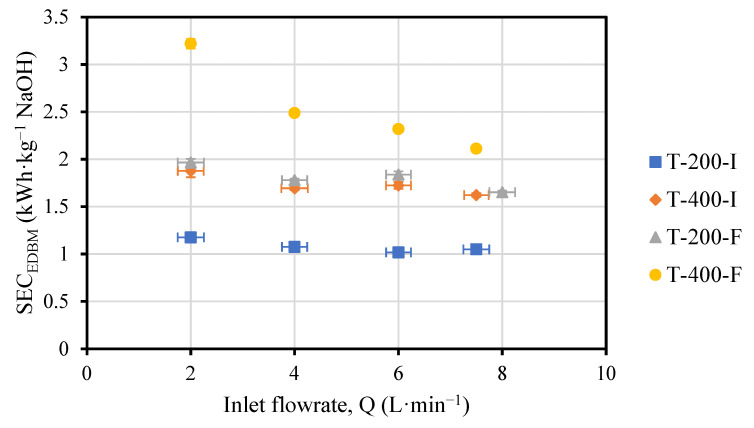
Values of SEC_EDBM_ (kWh·kg^−1^ NaOH) versus flowrate. Vertical error bars have been estimated via error propagation theory and are so low (i.e., <5%) that cannot be visualized in the figure.

**Figure 5 membranes-13-00200-f005:**
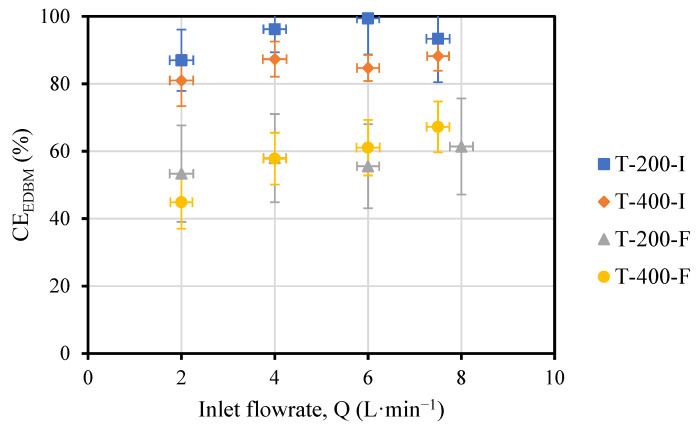
Profiles of the CE_EDBM_ versus the flowrate for tests.

**Figure 6 membranes-13-00200-f006:**
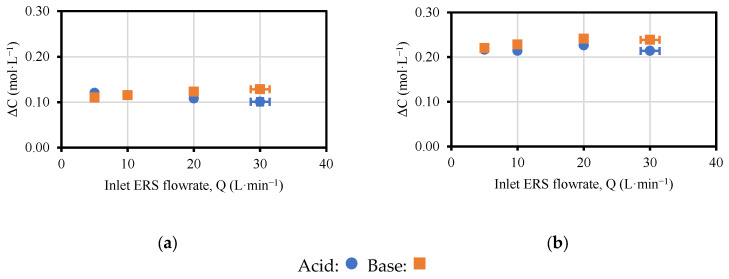
Values of concentration variation versus flowrate: (**a**) T-200-I-ERS; (**b**) T-400-I-ERS.

**Figure 7 membranes-13-00200-f007:**
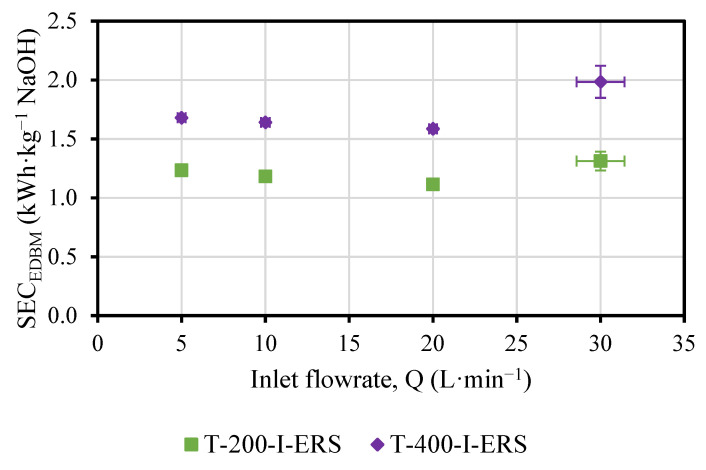
SEC_EDBM_ versus the flowrate (Q) and as a function of the applied current density.

**Figure 8 membranes-13-00200-f008:**
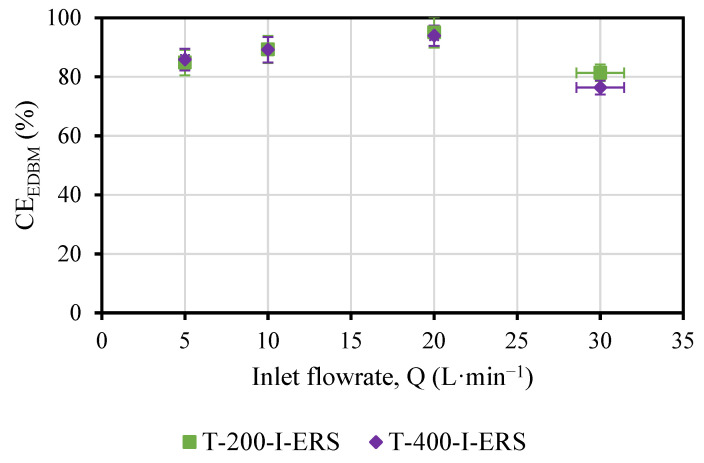
CE_EDBM_ versus the flowrate and as a function of the applied current density.

**Figure 9 membranes-13-00200-f009:**
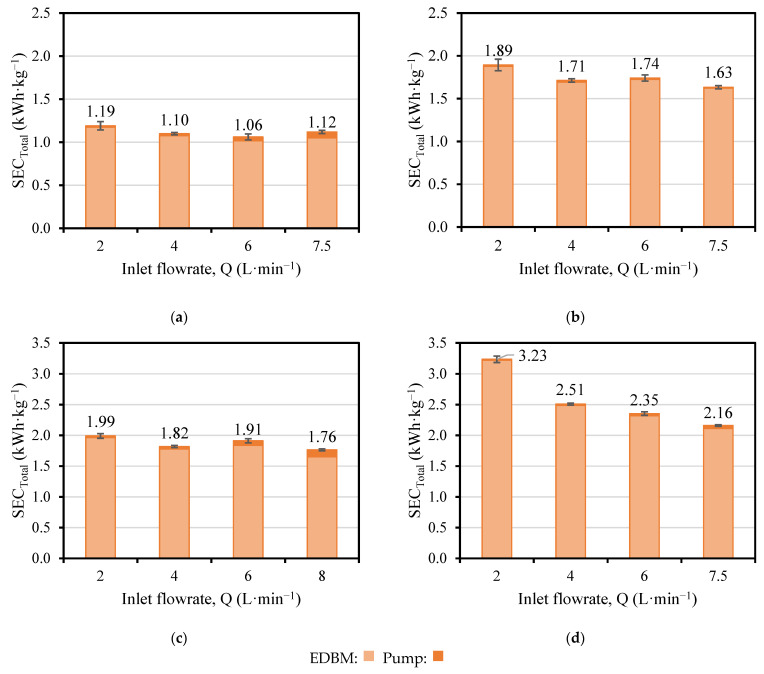
Values of SEC_Total_ versus flowrate: (**a**) T-200-I; (**b**) T-400-I; (**c**) T-200-F; (**d**) T-400-F.

**Figure 10 membranes-13-00200-f010:**
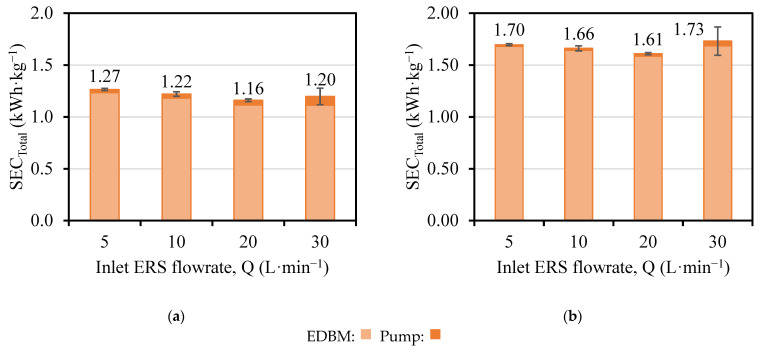
SEC_Total_ versus the flowrate and as a function of the applied current density (**a**) T-200-I-ERS; (**b**) T-400-I-ERS.

**Table 1 membranes-13-00200-t001:** Main properties of the FuMa-Tech BWT GmbH membranes used [30].

Membrane	FAB-PK-130	FKB-PK-130	FBM
Membrane type	AEM	CEM	BPM
Reinforcement	PEEK woven web	PEEK woven web	PEEK woven web
Thickness (µm)	130	130	150
Resistance (mΩ·cm^2^) ^1^	8	6.5	110
Selectivity (%) ^1^	>95	>97	>95
Swelling at 80 °C per dimension (%)	<4	< 4	<5
E-Modulus (MPa)	>1.500	>1.500	>1.500

^1^ Measured in 1 mol·L^−1^ NaCl solutions.

**Table 2 membranes-13-00200-t002:** Summary of the experimental conditions of the performed tests.

Code	Current Density (A·m^−2^)	Chamber	Concentrations (mol·L^−1^)	Flowrate (L·min^−1^)	Chamber Flow Velocity (cm·s^−1^)
T-200-I	200	Acid (HCl)	0.1	2.0–7.5	0.88–3.30
		Base (NaOH)	0.1	2.0–7.5	0.88–3.30
		Saline (NaCl)	1.0	2.0–7.5	0.88–3.30
		ERS (Na_2_SO_4_)	0.25	20	4.40
T-400-I	400	Acid (HCl)	0.1	2.0–7.5	0.88–3.30
		Base (NaOH)	0.1	2.0–7.5	0.88–3.30
		Saline (NaCl)	1.0	2.0–7.5	0.88–3.30
		ERS (Na_2_SO_4_)	0.25	20	4.40
T-200-F	200	Acid (HCl)	0.7	2.0–8.0	0.88–3.52
		Base (NaOH)	0.7	2.0–8.0	0.88–3.52
		Saline (NaCl)	0.5	2.0–8.0	0.88–3.52
		ERS (Na_2_SO_4_)	0.25	20	4.40
T-400-F	400	Acid (HCl)	0.7	2.0–7.5	0.88–3.30
		Base (NaOH)	0.7	2.0–7.5	0.88–3.30
		Saline (NaCl)	0.5	2.0–7.5	0.88–3.30
		ERS (Na_2_SO_4_)	0.25	20	4.40
T-200-I-ERS	200	Acid (HCl)	0.1	6.0	2.64
		Base (NaOH)	0.1	6.0	2.64
		Saline (NaCl)	1.0	6.0	2.64
		ERS (Na_2_SO_4_)	0.25	5.0–30	1.10–6.60
T-400-I-ERS	400	Acid (HCl)	0.1	6.0	2.64
		Base (NaOH)	0.1	6.0	2.64
		Saline (NaCl)	1.0	6.0	2.64
		ERS (Na_2_SO_4_)	0.25	5.0–30	1.10–6.60

All tests were carried out at ambient temperature (30 ± 2 °C).

## Data Availability

Data will be available by request.

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
