# Peer review of "Analysis of Operational Parameters in Acid and Base Production Using an Electrodialysis with Bipolar Membranes Pilot Plant"

_membranes, 2023, doi:10.3390/membranes13020200_

Round 1

Reviewer 1 Report

This study investigates acid and base production using electrodialysis with a bipolar membrane pilot plant. The authors examined the effect of process parameters, i.e., current density and flow rate, on the current efficiency and specific energy consumption for producing 1 kg of NaOH. Moreover, the authors analyzed the effect of electrolyte solution flow rate on the performance of the EDBM unit.

In my opinion, the manuscript was good-written and organized. However, several points should be clarified and improved the quality of the manuscript before accepting for publication.

Below, I present my comments, questions, and suggestions that may strengthen the scientific quality of the paper.

1.           Introduction

Line 119 – Please change the sentence “… can dissociate salts into acids and bases” to “…can convert…”.

-             Scientific novelty should be emphasized

2.2. Experiments design

Line 219 - On what basis were the current density values chosen? Was the limiting current density determined before the target experiment? Please complete the.

Table 1 - Most of the experiments were performed in the flow rate range of 2 - 7.5 L/min while the table shows the range of 2 – 8 L/min. Please explain.

2.3. Experimental procedure

Line 227 - Please explain why the initial volume of solution in the salt chamber was the same as the volume of solution in the acid and alkali chamber. By controlling the initial volume of the feed solutions, it is possible to concentrate the main product. In addition, please provide information on the change in the volume of working solutions (an effect typical for the EDBM process).

3.1. Effect of acid, base, and saline flowrates on the performance of the EDBM unit

-Fig. 3 Surprisingly, in the entire work, only for the processes coded as T-200-F, the upper range of the flow rate value was equal to 8.

-In Fig. 3, the error bars for the concentration change values are missing. Please explain.

- Fig. 3 Please indicate the number of repetitions represented by the error bars shown.

- Fig. 4 No error bars for variant coded as T-400-F.

3.3. Evaluation of pumping contribution to SECTotal

- Fig. 9a - No superscript for SEC Total

- Fig. 9 ac - The differences presented in Fig. 9 (a) and (c) are at the level of statistical error. It seems that for processes carried out at a lower current density of 200 A/m2, the flow rate of the electrolyte solution does not affect the efficiency of the EDBM process.

Overall information

In my opinion, the work could be enriched with a basic techno-economic analysis.

Reviewer 2 Report

The authors presented a paper entitled “Analysis of operational parameters in acid and base production using an electrodialysis with bipolar membranes pilot plant”.

Bellow, some comments which should be reported.

1. Introduction

-The introduction is so long.

-Abbreviation list is recommended.

Regarding the wastewater valorization streams, I recommend adding information across ZLD processes including diffusion dialysis (inorganic acids purification). Moreover, ED can be also used for the concentration/production of NaOH in different concentrations.

2. Material and Methods

Line 188: Please, provide supplier information (city).

Line 189: What is the type of IEX membranes (homogenous or heterogeneous)?

Line 190: Please, provide supplier information (city and country).

Lines 203-207: How the samples were collected and stored?

Line 224: Table 1. Instead of Channel can be used Chamber. Please, provide a linear speed for each compartment.

What are the type of distributors and the size used?

Electrical conductivity and pH during the EDBM process should be presented.

3. Results and Discussions

First of all, missing important information regarding membranes. The stability of the membranes should be confirmed by electrochemical or other types of analysis and discussed.

Lines 296-297: Should be discussed.

Lines 298-303: Please provide numbers (current density). The authors provide calculations but conditions are important as well. Limiting or over-limiting current and conditions should be discussed.    

Line 391: Figure 9. Should be min−1 / kg−1.

Round 2

Reviewer 1 Report

I accept the manuscript as it stands.

Reviewer 2 Report

The article can be accepted in its present form.